# The subset of peroxisomal tail-anchored proteins do not reach peroxisomes via ER, instead mitochondria can be involved

Tamara Somborac[1]*, Güleycan Lutfullahoglu Bal[1], Kaneez Fatima[1], Helena Vihinen[2], Anja Paatero[1], Eija Jokitalo[2], Ville O. Paavilainen[1], Svetlana Konovalova[1]*

**1** HiLIFE, Institute of Biotechnology, University of Helsinki, Helsinki, Finland, **2** Electron Microscopy Unit, Institute of Biotechnology, University of Helsinki, Helsinki, Finland

\* svetlana.konovalova@helsinki.fi (SK); tamara.somborac@helsinki.fi (TS)

## Abstract

Peroxisomes are membrane-enclosed organelles with important roles in fatty acid breakdown, bile acid synthesis and biosynthesis of sterols and ether lipids. Defects in peroxisomes result in severe genetic diseases, such as Zellweger syndrome and neonatal adrenoleukodystrophy. However, many aspects of peroxisomal biogenesis are not well understood. Here we investigated delivery of tail-anchored (TA) proteins to peroxisomes in mammalian cells. Using glycosylation assays we showed that peroxisomal TA proteins do not enter the endoplasmic reticulum (ER) in both wild type (WT) and peroxisome-lacking cells. We observed that in cells lacking the essential peroxisome biogenesis factor, PEX19, peroxisomal TA proteins localize mainly to mitochondria. Finally, to investigate peroxisomal TA protein targeting in cells with fully functional peroxisomes we used a proximity biotinylation approach. We showed that while ER-targeted TA construct was exclusively inserted into the ER, peroxisome-targeted TA construct was inserted to both peroxisomes and mitochondria. Thus, in contrast to previous studies, our data suggest that some peroxisomal TA proteins do not insert to the ER prior to their delivery to peroxisomes, instead, mitochondria can be involved.

## Introduction

Peroxisomes are single membrane-bounded organelles with essential metabolic functions in human cells. Peroxisomes are crucial in lipid metabolism, catabolism of very long chain fatty acids, biosynthesis of sterols, and ether lipids. Defects in peroxisome biogenesis result in critical genetic diseases, such as Zellweger syndrome, neonatal adrenoleukodystrophy, infantile Refsum disease, and myelopathies [1]. *De novo* peroxisomal biogenesis includes assembly of peroxisomal membrane proteins and subsequent import of soluble proteins into peroxisomal matrix [2, 3]. Although the main components and pathways of peroxisomal matrix protein import have been successfully identified [4–6], how peroxisomal membrane proteins (PMPs) are targeted to their respective destinations remains unclear [7, 8].

**Data Availability Statement:** All relevant data are within the paper and its Supporting Information files.

**Funding:** We express our sincere appreciation for the grant support extended to Cory Dunn for this study. The funding sources include the Academy of Finland (Grant Number 331556, https://www.aka.fi), Sigrid Jusèlius Foundation (https://www.sigridjuselius.fi), Jane ja Aatos Erkon Säätiö (Grant Number 200057, https://jaes.fi/), and HORIZON EUROPE European Research Council (Grant Number 637649, https://erc.europa.eu/). The open access of this publication was funded by the Helsinki University Library. The funders had no role in study design, data collection and analysis, decision to publish, or preparation of the manuscript.

**Competing interests:** The authors have declared that no competing interests exist.

A small portion of PMPs is embedded in the peroxisomal membrane by their C-terminal transmembrane segment, termed the tail-anchor (TA) that can direct newly synthesized TA proteins to different subcellular compartments. A combination of hydrophobicity of transmembrane domain and tail charge of TA proteins plays a role in the specific targeting of TA proteins to peroxisomes [9]. TA proteins carry out important functions in cell metabolism [10] and several disorders are linked to defective targeting of these proteins [11, 12]. Studies on PMP targeting in mammalian cells have suggested different modes of delivery to peroxisomes: a direct delivery from cytosol to pre-existing organelles or/and ER dependent trafficking. In the direct delivery model, PMPs reach peroxisomes without ER involvement by using essential peroxisome biogenesis factors [13, 14]. On the other hand, the indirect pathway suggests that PMPs are initially inserted into the ER membrane before being sorted to peroxisomes. It has been demonstrated that in the absence of mature peroxisomes many PMPs accumulate at the ER membrane [15, 16]. One study suggests that peroxisomes can be formed from PMPs deriving from both the ER and mitochondria [8].

Several distinct mechanisms of PMP delivery were proposed. One such mechanism implicates the role of the Sec61 translocon [17], a protein-complex channel involved in protein translocation across the ER membrane. Despite the diversity of proposed mechanisms for PMP delivery, a consistent element across all these models is the involvement of a specific set of proteins known as peroxins. These peroxins are indispensable in facilitating the trafficking of TA proteins through the ER.

Two highly conserved peroxins PEX19 and PEX3, are thought to be the key elements of PMP trafficking in cells [18]. While PEX19 is mainly a cytosolic chaperone, PEX3 acts as a membrane docking factor for protein insertion [10]. Deletion of one of these peroxins results in cells devoid of mature peroxisomes.

The process by which peroxisomal TA proteins in mammals are targeted to peroxisomes is a critical element for our understanding of peroxisomal cell biology. However, the targeting mechanisms of mammalian peroxisomal TA proteins remain largely unexplored.

We studied the pathway of TA proteins targeting peroxisomes using biochemical assays and microscopy. We took advantage of YgiM(TA), a well-characterized bacteria-derived TA protein targeted specifically to peroxisomes in mammalian cells [19]. Its small size and lack of function-directed interactions within eukaryotic cells make it a versatile tool to investigate protein targeting to peroxisomal membrane. Our data suggest that peroxisomal TA proteins do not insert ER prior to their delivery to peroxisomes. Instead, mitochondria can play a role in the targeting of TA proteins to peroxisomes.

## Results

### Exogenous peroxisomal TA protein does not transit through the ER in mammalian cells

We rationalized that if peroxisomal TA proteins become exposed to the ER lumen at some point during the delivery to mature peroxisomes, then in the absence of peroxisomes, peroxisomal TA proteins could be detectable at the ER. Therefore, we generated HEK293T cell lines lacking peroxisomes by deletion of the essential peroxisomal biogenesis factors, PEX3 or PEX19, using CRISPR Cas9 editing. Immunoblotting of the resulting cell lines showed the specific deletion of either PEX3 or PEX19 (Fig 1A). As expected from previous research [13, 14], confocal microscopy revealed lack of peroxisomes in both cell lines as assessed by immunofluorescence labeling of catalase (Fig 1B).

To test whether peroxisomal TA proteins insert into the ER on their way to peroxisomes, we used a previously described model protein YgiM(TA), a bacterial derived sequence, shown

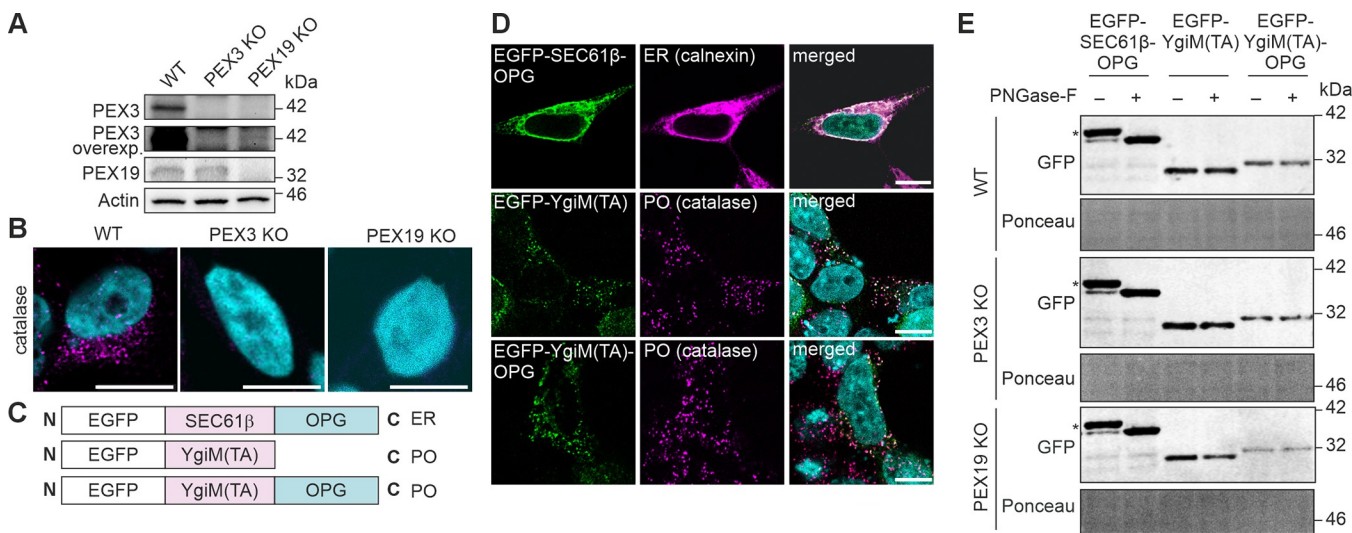

**Fig 1. Exogenous peroxisomal TA protein does not transit through the ER in mammalian cells.** (A) Immunoblotting analysis of PEX3 KO or PEX19 KO HEK293T cells generated by CRISPR Cas9 approach. WT, wild type, parental HEK293T cells were used as a control. Actin was used as a loading control. (B) Confocal immunofluorescence imaging showed the absence of peroxisomes in PEX3 KO or PEX19 KO HEK293T cells. WT, wild type, parental HEK293T cells were used as a control. Anti-catalase antibody was used to visualize peroxisomes (magenta). DAPI (cyan) shows nuclei. Scale bars, 10 μm. (C) Schematic view of the OPG-fused constructs used in glycosylation assays. EGFP-YgiM(TA) without OPG was used as a control. (D) Intracellular localization of the constructs used in glycosylation assay analyzed by confocal immunofluorescence imaging. WT HEK293T were transiently transfected with EGFP-SEC61β-OPG, EGFP-YgiM(TA) or EGFP-YgiM(TA)-OPG constructs for 24 h. Anti-GFP antibody was used to visualize the EGFP constructs (green). Calnexin was used to visualize ER (magenta), catalase was used to visualize peroxisomes (PO, magenta). DAPI (cyan) shows nuclei. Scale bars, 10 μm. (E) TA substrates with or without OPG tag were transfected to WT, PEX3 KO or PEX19 KO HEK293T cells. Cell lysates were analyzed by immunoblotting using anti-GFP antibody. Glycosylation bands are indicated by asterisk. Glycosylation was confirmed by loss of the band after addition of glycosidase, PNGase-F. Ponceau staining was used as a loading control.

to specifically target peroxisomes in mammalian cells [20]. As a reporter for protein entry into the ER we used opsin glycosylation tag (OPG) since it contains a glycosylation site which accepts glycans if exposed to the ER lumen [21]. OPG was fused to the C-terminus of YgiM (TA), a peroxisomal TA construct, or SEC61β, a known ER-targeted TA construct (Fig 1C). Confocal microscopy coupled with specific immunostaining demonstrated that the OPG tag did not influence localization of either construct (Fig 1D), consistent with the previous work on C-terminally inserted OPG tag in TA proteins [22, 23].

WT, PEX3 KO or PEX19 KO HEK293T cells were transiently transfected with peroxisomal TA protein, YgiM(TA)-OPG, or ER TA protein, SEC61β-OPG. SEC61β-OPG expressed in control cells or cells lacking peroxisomes showed clear glycosylation as detected by a western blot band shift when treated with PNGase-F that removes oligosaccharides from glycoproteins (Fig 1E). In contrast, YgiM(TA)-OPG glycosylation was non-detectable in control cells or in cells lacking peroxisomes. Together, these results suggest that peroxisomal TA proteins do not enter ER before their insertion into the peroxisome. Also, peroxisomal TA proteins do not appear to target the ER even in the absence of mature peroxisomes (Fig 1E).

## Exogenous YgiM(TA) and endogenous ACBD5 peroxisomal TA proteins are targeted to mitochondria in the absence of peroxisomes

Since peroxisomal TA proteins do not appear to enter ER in control cells or cells lacking peroxisomes, we sought to identify the localization of peroxisomal TA proteins in cells devoid of peroxisomes. For this purpose, we generated cells stably expressing peroxisomal TA protein, YgiM(TA), in WT, PEX3 KO or PEX19 KO HEK293T cells. We analyzed the colocalization of

YgiM(TA) with peroxisomes, ER or mitochondria by immunofluorescence microscopy and quantitative colocalization analysis. In WT cells YgiM(TA) localized strongly to peroxisomes (Fig 2A–2C). In PEX3 KO or PEX19 KO cells YgiM(TA) showed colocalization with both ER and mitochondria, with most notable colocalization with mitochondria (Fig 2C).

To visualize YgiM(TA) localization at superior resolution level, we used immunoelectron microscopy. Immunogold electron microscopy analysis showed that while in WT cells YgiM (TA) localized mainly to peroxisomes, in the absence of peroxisomes it was primarily targeting mitochondria (Fig 2D). We analyzed electron microscopy images using stereological analysis to identify mitochondrial relative labeling index [24, 25]. Consistent with the immunofluorescence experiments (Fig 2A–2C), we observed robust localization of YgiM(TA) to mitochondria in PEX3 KO or PEX19 KO cells (Fig 2E).

YgiM(TA) is an exogenous peroxisomal TA protein that was overexpressed in HEK293T cells in our model. Since overexpressed proteins have the propensity to mistarget [26], we analyzed localization of an endogenous peroxisomal TA protein, ACBD5, in cells lacking peroxisomes. By immunofluorescent microscopy and quantitative colocalization analysis we showed that ACBD5 is strictly localized to peroxisomes in WT HEK293T cells (Fig 3A and 3D). In PEX19 KO cells ACBD5 was detected in mitochondria, but not in the ER (Fig 3B–3D). Notably, ACBD5 was barely detectable by immunofluorescent analysis or by immunoblotting in PEX3 KO cell (Fig 3A–3C and 3E), suggesting its degradation. Thus, the endogenous peroxisomal TA protein localized to mitochondria in the absence of peroxisomes, suggesting that mitochondria, but not ER may be involved in peroxisomal targeting of TA proteins in mammals.

## Exogenous peroxisomal TA protein inserts to mitochondria in mammalian cells with intact peroxisomes

Having established that in the absence of peroxisomes, the peroxisomal TA proteins can localize to mitochondria, we sought to investigate the route by which they reach peroxisomes under normal conditions. To achieve this, we used a proximity biotinylation approach. We used wild type BirA/AviTag labeling approach [27] to reveal interactions between peroxisomal TA (AviTagged) and specific organelles containing a localized BirA protein. This approach is based on the ability of a biotin ligase (BirA) to biotinylate AviTag in a highly specific manner only if both tagged proteins are present in the same cellular compartment [28, 29]. We introduced a doxycycline-inducible BirA construct targeting ER (ER-BirA), mitochondrial intermembrane space (IMS-BirA) or peroxisomes (PO-BirA) in the Flp-In T-Rex™ HEK293 cells using corresponding targeting sequences. We then fused the AviTag sequence to the C-terminus of EGFP-tagged ER or peroxisomal TA protein (Fig 4A and 4B). These AviTag constructs were subsequently transiently transfected to the Flp-In T-Rex™ HEK293 cells stably expressing different organelle-targeted BirA constructs. To avoid overexpression of the AviTag constructs, we used the weak promoter, UBC [30]. We then confirmed the expected organelle localization of BirA and AviTag constructs using immunofluorescence microscopy (Fig 4C and 4D).

To ensure the specificity of the assay we kept the expression of BirA at a very low level by using inherent leakiness in our doxycycline system without adding additional doxycycline to the culture media. Flp-In T-Rex™ HEK293 cells expressing ER-BirA, IMS-BirA or PO-BirA were transiently transfected with ER-targeted TA construct, SEC61β-AviTag, or with peroxisome-targeted TA construct, YgiM(TA)-AviTag. Using streptavidin immunoblotting we showed that as expected SEC61β-AviTag was biotinylated in ER-BirA cells but not in IMS-BirA or PO-BirA expressing cells (Fig 4E), validating the assay as a reliable tool to analyze

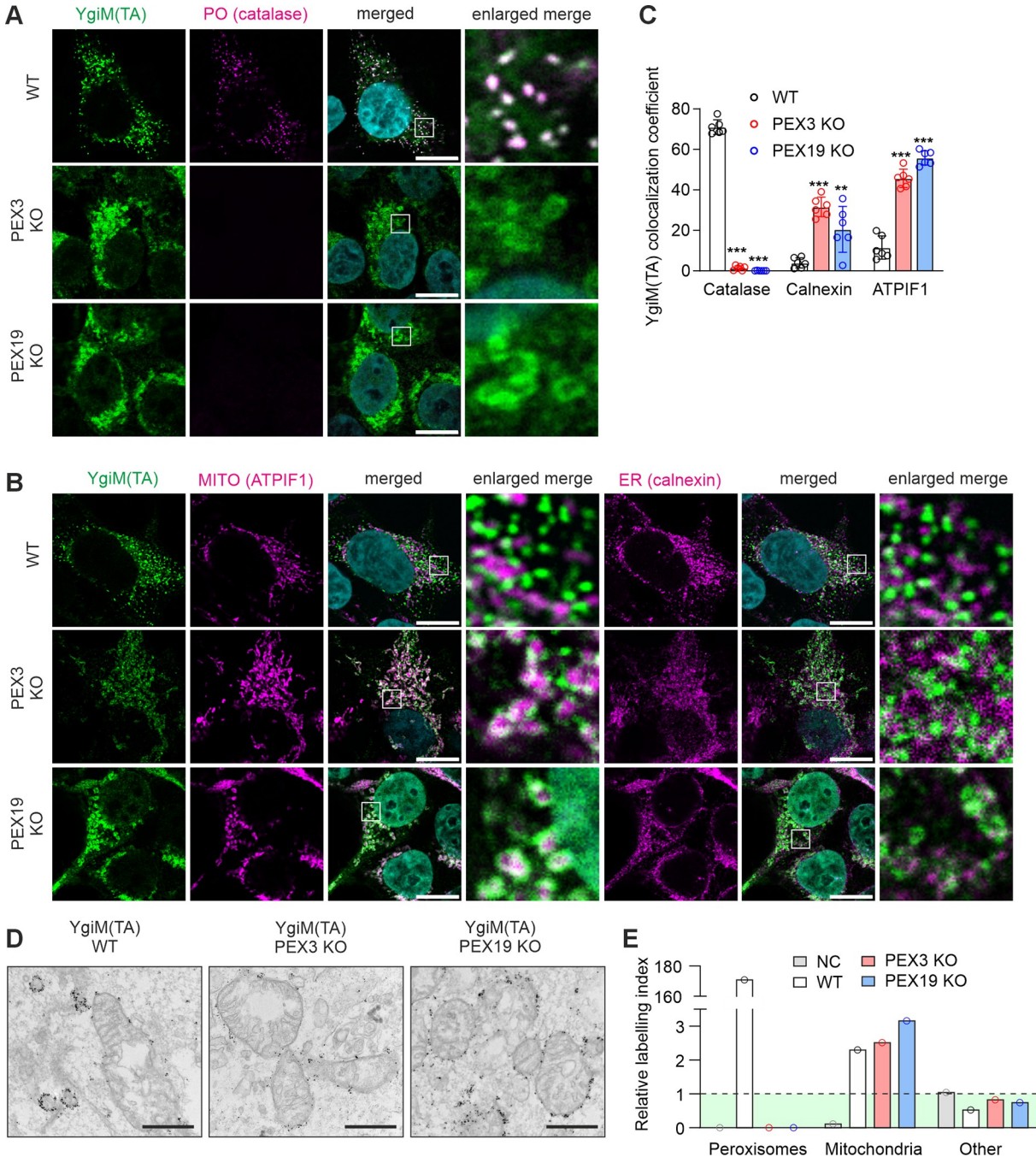

**Fig 2. Exogenous peroxisomal TA YgiM(TA) is targeted to mitochondria in mammalian cells lacking peroxisomes.** (A and B) Localization of EGFP-YgiM(TA) stably expressed in WT, PEX3 KO or PEX19 KO HEK293T cells. (A) Confocal immunofluorescence imaging showing EGFP-YgiM(TA) localization. Anti-GFP antibody was used to visualize EGFP-YgiM(TA) (green). Anti-catalase antibody was used to label peroxisomes (PO, magenta). DAPI (cyan) shows nuclei. Scale bars, 10 μm. (B) Confocal immunofluorescence imaging showing EGFP-YgiM (TA) localization. Anti-GFP antibody was used to visualize EGFP-YgiM(TA) (green). Anti-ATPIF1 antibody was used to label mitochondria (MITO, magenta) and anti-calnexin antibody was used to label ER (magenta). DAPI (cyan) shows nuclei. Scale bars, 10 μm. (C) Colocalization of EGFP-YgiM(TA) with catalase, calnexin or ATPIF1 was analyzed by Mander's overlap coefficient using confocal microscopy images. For each sample 5–6 fields of view were analyzed, each view captured approximately 50 cells. The data are presented as mean ± SD. **$P < 0.01$, ***$P < 0.001$ as compared to WT cells (unpaired t-tests, n = 5–6). (D) Electron microscopy imaging of WT, PEX3 KO or PEX19 KO HEK293T cells stably expressing EGFP-YgiM(TA). Anti-GFP immunogold staining was used to visualize EGFP-YgiM(TA). Scale bars, 500 nm. (E) Stereological analysis of electron microscopy images was used to identify relative labeling index of YgiM(TA) in peroxisomes or mitochondria. If no association with mitochondria or peroxisomes was observed, the gold particle label was assigned to the 'other' category. The 'other' category includes all other organelles (except mitochondria and peroxisomes) and cytosol. NC, negative control, cells not expressing

EGFP-YgiM(TA) construct used for quantitation of background signal from immunogold labeling. Area above the green surface depicts non-randomly distributed gold particles. For each cell line 15 cells were analyzed (2 images per cell with the area of 300 µm²).

the route of targeting of TA proteins that are ultimately delivered to different target organelles. In contrast, YgiM(TA)-AviTag biotinylation was exclusively detected in cells expressing PO-BirA or IMS-BirA (Fig 4E). Therefore, we conclude that peroxisomal TA proteins pass through mitochondria, but not through ER, in cells containing intact mature peroxisomes.

Collectively, our results showed that peroxisomal TA proteins may target mitochondria in mammalian cells both in the presence and in absence of peroxisomes, suggesting that mitochondria could be involved in the targeting of peroxisomal TA proteins in mammalian cells.

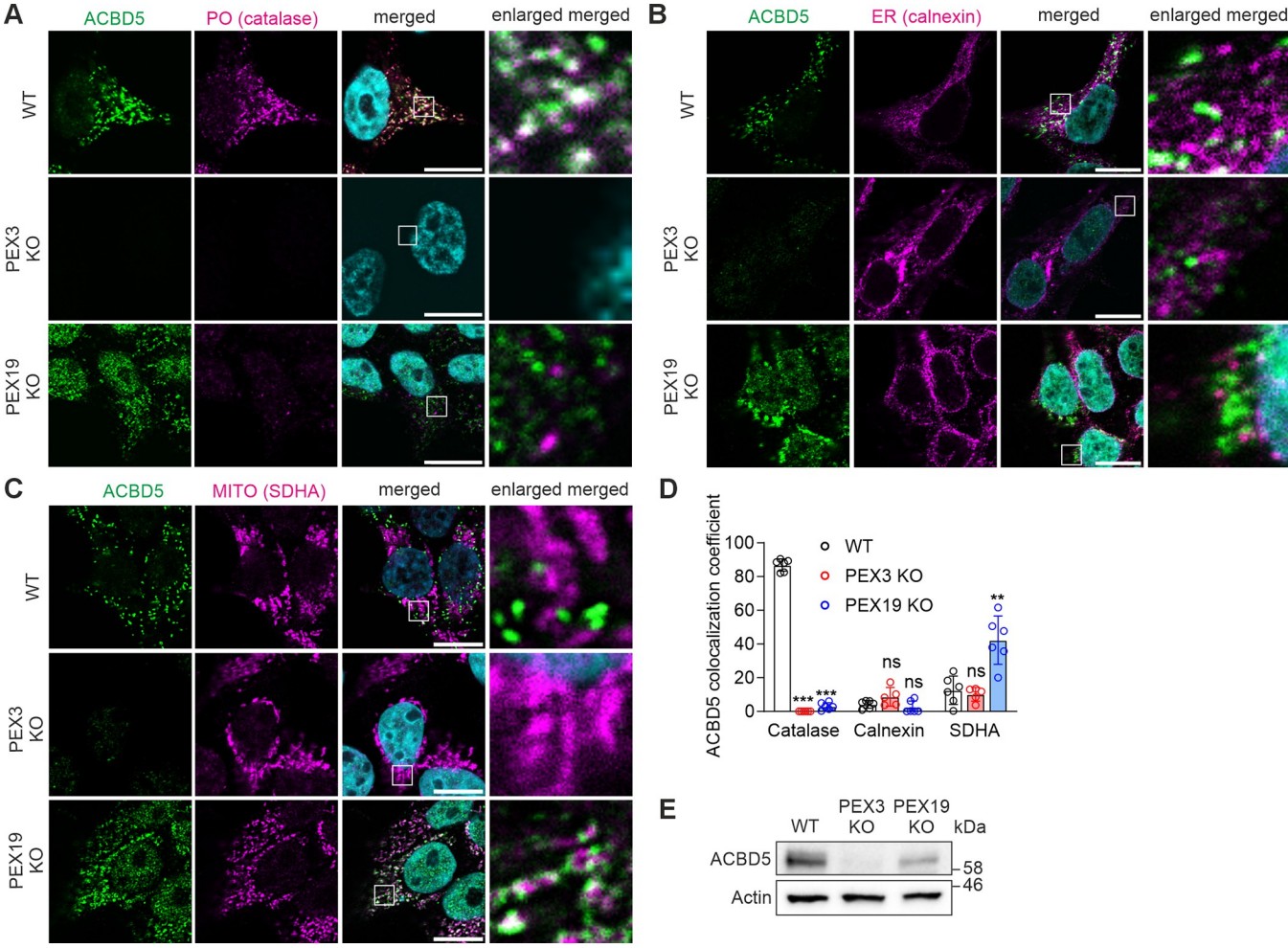

**Fig 3. Endogenous peroxisomal TA protein ACBD5 localizes to mitochondria in the absence of peroxisomes in mammalian cells.** (A–C) Localization of ACBD5 in WT, PEX3 KO or PEX19 KO HEK293T cells. (A) Confocal immunofluorescence imaging showing ACBD5 localization. Anti-ACBD5 antibody was used to visualize ACBD5 (green), anti-catalase antibody was used to label peroxisomes (PO, magenta). DAPI (cyan) shows nuclei. Scale bars, 10 µm. (B) Confocal immunofluorescence imaging showing ACBD5 localization. Anti-ACBD5 antibody was used to visualize ACBD5 (green), anti-calnexin antibody was used to label ER (magenta). DAPI (cyan) shows nuclei. Scale bars, 10 µm. (C) Confocal immunofluorescence imaging showing ACBD5 localization. Anti-ACBD5 antibody was used to visualize ACBD5 (green), anti-SDHA antibody was used to label mitochondria (MITO, magenta). DAPI (cyan) shows nuclei. Scale bars, 10 µm. (D) Colocalization of ACBD5 with catalase, calnexin or SDHA was analyzed by Mander's overlap coefficient using confocal microscopy images. For each sample 5–6 fields of view were analyzed, each view captured approximately 50 cells. The data are presented as mean ± SD. **$P < 0.01$, ***$P < 0.001$, ns, not significant as compared to WT cells (unpaired t-tests, n = 5–6). (E) Immunoblotting analysis of ACBD5 in WT, PEX3 KO or PEX19 KO HEK293T cells. Actin was used as a loading control.

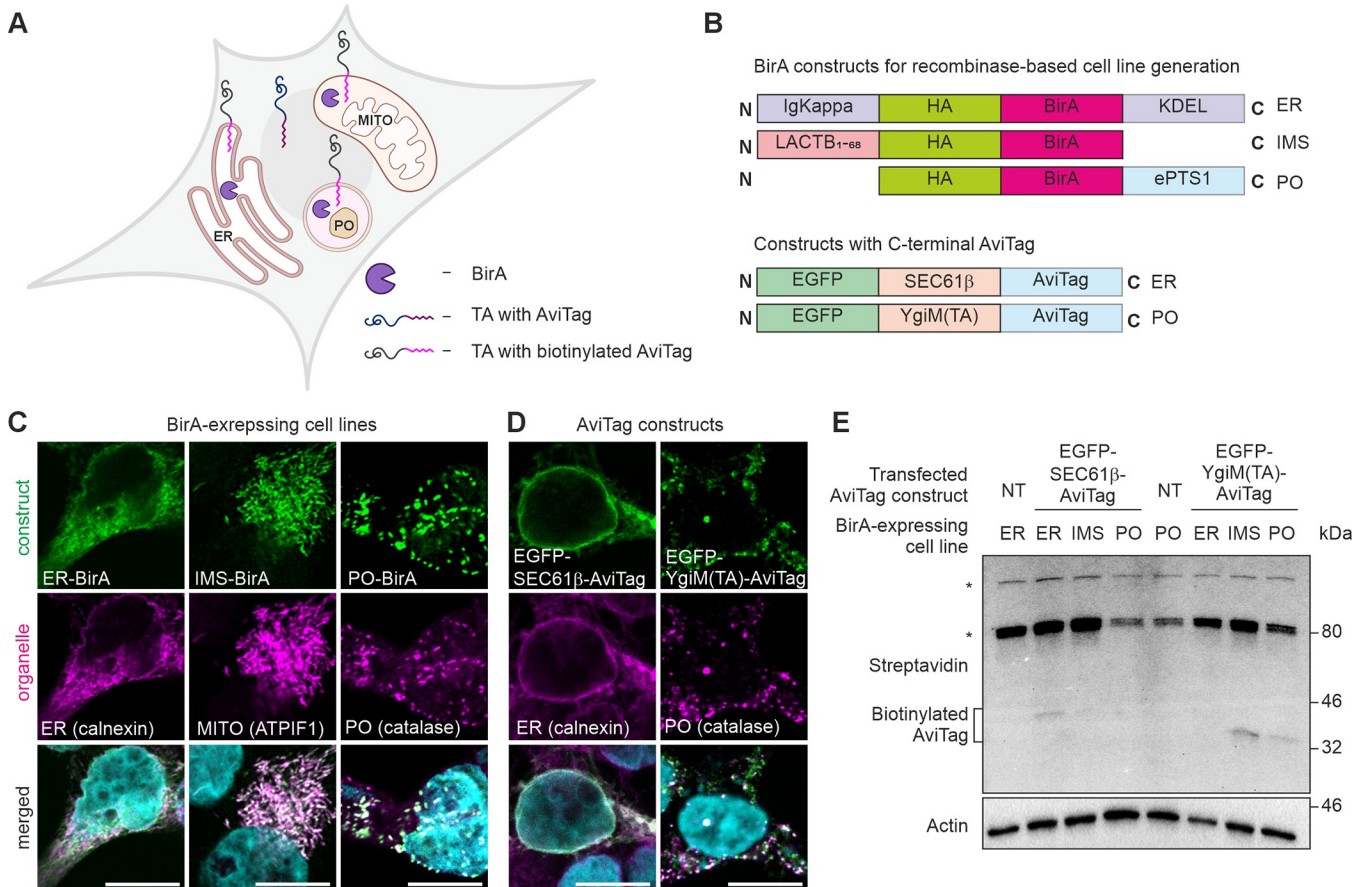

**Fig 4. Exogenous peroxisomal TA protein inserts to mitochondria in mammalian cells with intact peroxisomes.** (A) Schematic representation of the biotinylation experiment. TA constructs fused to AviTag are transfected to the cells expressing BirA in endoplasmic reticulum (ER), peroxisomes (PO) or mitochondria (MITO). Once TA inserts to the specific organelle labeled with BirA, biotin acceptor sequence of the TA, AviTag, is biotinylated by BirA. (B) Schematic view of the constructs used to generate BirA-expressing cell lines and TA constructs fused to AviTag. (C) Doxycycline-inducible BirA constructs targeted to ER (ER-BirA), mitochondrial intramembrane space (IMS-BirA) or peroxisomes (PO-BirA) were stably introduced into Flp-In™ T-REx™-293 cells. The expression of the constructs was induced by adding of 10 μg / mL doxycycline for 24 h. The intracellular localization of doxycycline-inducible BirA constructs targeted to the specific organelle was analyzed by confocal immunofluorescence imaging. Anti-BirA antibody was used to visualize BirA construct (green), anti-calnexin antibody was used to label ER (magenta), anti-ATPIF1 antibody was used to visualize mitochondria (magenta), anti-catalase antibody was used to label peroxisomes (PO, magenta). DAPI (cyan) shows nuclei. Scale bars, 10 μm. (D) Flp-In™ T-Rex™-293 cells were transiently transfected with EGFP-SEC61β-AviTag or EGFP-YgiM(TA)-AviTag for 24 h. The intracellular localization of AviTag constructs was analyzed by confocal immunofluorescence imaging. Anti-GFP antibody was used to visualize AviTag constructs (green), anti-calnexin antibody was used to label ER (magenta), anti-catalase antibody was used to label peroxisomes (PO, magenta). DAPI (cyan) shows nuclei. Scale bars, 10 μm. (E) Flp-In T-Rex™ HEK293 cells expressing ER-BirA, IMS-BirA or PO-BirA were transiently transfected with ER-targeted TA construct, SEC61β-AviTag, or with peroxisome-targeted TA construct, YgiM(TA)-AviTag. Cell lysates were analyzed by immunoblotting using streptavidin. Bands of ~33 kDa and 39 kDa indicate biotinylated AviTag constructs. Endogenous biotinylated mitochondrial proteins are shown by asterisks. Actin was used as a loading control. NT, non-transfected control. Fig (A) was created with BioRender.com.

## Discussion

Research on targeting pathways of peroxisomal TA in mammalian cells provided data both in favor [7] and against a route involving the ER [31, 32]. It has been demonstrated that in yeast [33, 34] and plants [35] peroxisomal TA proteins insert to the ER prior to reaching peroxisome. Further, in yeast cells devoid of peroxisomes, peroxisomal TA proteins were visualized in structures described as pre-peroxisomal vesicles derived from ER [20, 36–38]. Using glycosylation assays in human cells, we tested the possibility of peroxisomal TA transiting through the ER in both WT and cells lacking peroxisomes. To generate cells lacking peroxisomes we knocked out essential membrane or cytosolic peroxisomal biogenesis factors, PEX3 or PEX19,

respectively. The absence of N-linked glycosylation provided evidence against ER-dependent targeting of peroxisomal TA proteins in mammalian cells. However, we cannot fully exclude the possibility that peroxisomal TA proteins might transiently visit the ER lumen, without becoming glycosylated. Therefore, we analyzed the localization of peroxisomal TA proteins in mammalian cells by immunofluorescence and immunoelectron microscopy imaging. We observed that in the absence of peroxisomes in PEX19 KO cells, endogenous peroxisomal TA protein ACBD5 localizes mainly to mitochondria. Mitochondrial localization of peroxisomal membrane proteins in the absence of peroxisomes has been reported previously [39, 40], although, this finding was considered as a mistargeting artifact. However, we showed that in PEX3 KO cells, which lack peroxisomes as well, ACBD5 does not target mitochondria. These data suggest that mitochondrial localization of peroxisomal membrane proteins may not represent simple passive mistargeting, but instead an active process in which PEX3 may play an important role. Interestingly, PEX3 and PEX14 were found to be targeted to mitochondria prior to their assembly into peroxisomes (Sugiura et al) [8], suggesting the involvement of these factors in the insertion of peroxisomal TA proteins to mitochondria. It is also possible that PEX19 interacts with ACBD5 and prevents its mistargeting to mitochondria, thus in PEX19 KO cells we observed ACBD5 in mitochondria, while in PEX3 KO cells ACBD5 did not target mitochondria and was degraded in the cytosol.

We observed almost complete depletion of ACBD5 in PEX3 KO cells, suggesting that at least some untargeted peroxisomal TA proteins become directed for degradation. Notably, PEX19 KO causes reduction of PEX3 protein level, suggesting that PEX3 is degraded when peroxisomes are absent. However, the residual level of PEX3 in PEX19 KO cells was sufficient to protect ACBD5 from complete degradation in PEX19 KO cells. It is possible that PEX3 is involved in the transcriptional regulation of ACBD5 and modulates its expression. Deletion of PEX3 could disrupt this regulatory mechanism, resulting in the loss of ACBD5 expression. It was demonstrated that yeast orthologue of ACBD5, Atg37, binds PEX3 at the peroxisomal membrane [41] ensuring correct localization of PEX3 to peroxisomes [42]. PEX3 may influence the stability or post-translational modifications of ACBD5 such as proper folding, stabilization, or localization within the cell. Deletion of PEX3 may disrupt this interaction, leading to the degradation or mislocalization of ACBD5. Finally, PEX3 knockout may indirectly affect ACBD5 by disrupted cellular processes or signaling pathways that are necessary for the expression or stability of ACBD5.

Further investigations are warranted to elucidate the specific molecular mechanisms underlying the relationship between PEX3 and ACBD5. These studies could involve examining the transcriptional regulation of ACBD5 in PEX3-deficient cells, assessing the stability and post-translational modifications of ACBD5 in the absence of PEX3, and investigating potential downstream signaling pathways affected by PEX3 deletion that could influence ACBD5 expression.

Using organelle-specific proximity biotinylation assays we showed that peroxisomal TA protein targets both peroxisomes and mitochondria in cells containing intact peroxisomes. This indicates that mitochondria-related targeting may be a process that occurs during peroxisomal TA trafficking under normal conditions. Overall, our data suggest that TA proteins targeting to mitochondria, previously described as mislocalization, may be part of the endogenous targeting pathway to peroxisomes. These observations are consistent with the findings by Sugiura *et al*. [8] showing that newly assembled peroxisomes arise from both ER and the mitochondria.

Taken together, we observed a subset of YgiM(TA) localizing to the mitochondria in wild-type cells (Fig 2D), and subsequent biotinylation by a mitochondrial protein tagged with BirA (Fig 4E). These observations led us to the hypothesis that YgiM(TA), and potentially other

peroxisomal tail-anchored proteins, might transit via mitochondria *en route* to peroxisomes. However, it is crucial to consider alternative interpretations. One plausible scenario is that YgiM(TA) is dually localized to both peroxisomes and mitochondria. This would suggest that the localization of YgiM(TA) is not necessarily linear, but instead might follow parallel pathways, with minor subsets of the protein targeted independently to mitochondria. This concept of dual localization could provide additional complexity to understanding peroxisomal TA protein routing, supports previous data on dual localization of some mitochondrial TA proteins in yeast [43] and mammals [44], while also challenges our current view of organelle-specific peroxisomal TA protein targeting. It is still not clear why at least some TA proteins pass mitochondria *en route* to the peroxisome. Mitochondria and peroxisomes share several functional overlaps, including fatty acid and reactive oxygen species (ROS) metabolism [45, 46]. During mitochondrial transit, TA proteins potentially can undergo specific post-translational modifications or protein-protein interactions that prime them for their peroxisomal function. This transit could also serve as a quality control step, ensuring that only correctly folded and functional TA proteins reach the peroxisome. In addition, the functional interplay between peroxisomes, mitochondria, and ER [47–49] might necessitate certain proteins to be present in multiple organelles, leading to dual targeting.

We believe that our study offers valuable insights into the localization of peroxisomal TA proteins and invites further exploration of these complex cellular mechanisms.

## Materials and methods

### Cell culture

Human embryonic kidney cells (HEK293T) or Flp-In T-Rex™-293 cells (ThermoFisher, R780-07) were cultured (37˚C, 5% CO2) in DMEM (Pan-Biotech, P04-03600), supplemented with 10% fetal bovine serum (GIBCO, 10270106), l-glutamine (GIBCO, 25030081) and penicillin/streptomycin (GIBCO,15140122). TransIT-X2 (MirusBio, MIR 6000) was used for cell transfection according to manufacturer's instruction.

### Generation of knockout cell lines

CRISPR/Cas9 was used to generate knockout HEK293T cells. Cells were co-transfected with two gRNAs and pSpCas9n(BB)-2A-Puro (PX462) V2.0 (Addgene, 62987). PEX3 gRNAs were ACGTGCTTGAAAGGGGGCAT and CACCTCCAAGGACCGTGCCC. PEX19 gRNAs were GGGCCCCAGAAGAGATCGCC and GGGGCCGTGGTGGTAGAAGG. After 24 h the cells were selected with puromycin and expanded. T7 assay was used to confirm genome editing. Single cell clones were obtained and validated using immunoblotting analysis.

### Generation of stably expressing cell lines

HEK293T cells stably expressing EGFP-YgiM(TA) were generated by using standard lentiviral cell line generation protocol. Briefly, WT, PEX3 KO or PEX19 KO cells were co-transfected with pMD2.G (Addgene, 12259), psPAX2 (Addgene,12260) and pLL3.7-EGFP-YgiM(TA) using TransIT-2020 (MirusBio, MIR 5400) following manufacturer's instructions. Single cell clones were selected for low GFP expression, just above the base autofluorescence as measured in the WT cells by FACS.

Doxycycline-inducible BirA constructs were introduced into Flp-In™ T-Rex™-293 cells by co-transfecting with pOG44 (ThermoFischer, V600520) and corresponding BirA construct using TransIT-2020 (MirusBio, MIR5400) according to manufacturer's instructions. After 48 h the selection antibiotic was added (100 μg / ml hygromycin B (ENZO, ALX-380-306-G001)).

Every 48 h the growth medium was replaced with the fresh one, containing fresh selection antibiotic. The cells were under selection for 2.5 to 4 weeks until sufficient number of colonies were formed.

## Plasmid construction

OPG-tagged plasmids were constructed by homologous recombination in yeast as previously described [50]. Briefly, to generate pEGFP-SEC61β-OPG, mCherry-SEC61β was PCR amplified from mCh-Sec61 beta (a gift from Gia Voeltz (Addgene plasmid, 49155; http://n2t.net/addgene:49155; RRID:Addgene 49155)) with primers `ATGGTGAGCAAGGGCGAGGA` and `TTACCCTGTCTTATTGCTAAATGGAACGTAAAAGTTAGGACCCGAACGAGTGTACTTGCC CCAAATGTG`. SEC61β-OPG was generated by PCR amplification using primers `ATCACTC TCGGCATGGACGAGCTGTACAAGAGATCTATGCCTGGTCCGACC` and `GGTATGGCTGATTA TGATCAGTTATCTAGATTACCCTGTCTTATTGCTAAATGGAAC` and the obtained product was co-transformed with pEGFP-C1-2μ-URA3 cut with BamHI (NEB, R3136) into yeast strain BY4743 (EUROSCARF Y20000) cut. The obtained yeast plasmid DNA transformed into competent cells (NEB, C2987H).

To generate pEGFP-YgiM(TA)-OPG, YgiM(TA)-OPG sequence was produced by PCR from pRS315-mCherry-YgiM(TA) with primers `ATCACTCTCGGCATGGACGAGCTGTACA AGATGGTAGAGGATAAGATCCAGAAGGAAACA` and `TATGGCTGATTATGATCAGTTATCT AGATTACCCTGTCTTATTGCTAAATGGAACGTAAAAGTTAGGACCGTTCATCCAGCGATC TTTG`. The obtained PCR product co-transformed with pEGFP-C1-2μ-URA3, that was cut with BamHI (NEB, R3136), into yeast strain BY4743 (EUROSCARF Y20000). The obtained yeast plasmid DNA was transformed into competent cells (NEB, C2987H). Yeast strains, plasmids, and culture conditions are as described in [50].

To generate pLL3.7-EGFP-YgiM(TA), pLL3.7 (Addgene, 11795) was cut with EcoRI (NEB, R3101) and NheI (NEB, R3131) and the 6907 bp band was gel purified (Macherey-Nagel, 740609.50). EGFP-YgiM(TA) was PCR amplified from pEGFP-YgiM(TA) with primers `ATATTGCTAGCGCTACCGGTCGCCACCATG` and `CCTACTGAATTCTTAGTTCATCCAGC GATCTTTGC`. The obtained product was then cut with NheI (NEB, R3131) and EcoRI (NEB, R3101) and PCR purified (Macherey-Nagel, 740609.50). The two purified fragments were ligated using T4 DNA ligase (NEB, M0202) following manufacturer's instructions and transformed into competent cells (NEB, C2987H).

To generate BirA-constructs, pDisplay-BirA-ER (a gift from Alice Ting (Addgene plasmid # 20856; http://n2t.net/addgene:20856; RRID:Addgene_20856)) was used as a PCR template for ER-BirA or PO-BirA. The HA-BirA region was PCR amplified with primers `GCGAAGCTTTGGGGATATCCACCATGGAGACAGAC`/`GCGGGATCCGTTCGTCGACTCACAGC TCGTCCTT` for ER-BirA and `TTTTAAGCTTGCCACCATGTATCCATATGATGTTCC`/ `TTTTGGATCCTCATAGCTTACTTCTTCTGCCACGCCCCAGTTTTTCTGCACTAC` for PO-BirA. IMS-BirA (LACTB$_{1-68}$-HA-BirA) construct was purchased from Twist Bioscience (San Francisco, California, United States). The IMS sequence was as in [51]. The IMS-BirA construct and PCR products from ER-and PO-BirA products were cut with HindIII (NEB, R3104) and BamHI (NEB, R3136) and cloned into pcDNA™5/FRT/TO (TermoFischer, V652020) that was cut with the same restriction enzymes.

To generate AviTag constructs, pLJC5-3XHA-EGFP-PEX26 (Addgene, 139054) was cut with AgeI (NEB, R3552) and EcoRI(NEB, R3101). The large DNA fragment was gel purified (Macherey-Nagel, 740609.50) and used as a target backbone for cloning the AviTag constructs. AgeI-EGFP-SEC61β sequence and linker AviTag-EcoRI were purchased from Integrated DNA Technologies (IDT, Coralville, Iowa, USA), PCR amplified, annealed, ligated with the

cut backbone and transformed into competent cells (NEB, C2987H). EGFP-YgiM(TA) was PCR amplified from pEGFP-YgiM(TA)-noFis1(119–128) with appropriate primers, annealed with linker AviTag sequence as above and cloned to competent cells. GSGSTSGSGK was used as a linker sequence [27].

## Glycosylation assay in cells

Cells were transiently transfected with pEGFP- SEC61β-OPG or pEGFP-YgiM(TA)-OPG using TransIT2020 transfection reagent (MirusBio, MIR 5400) according to manufacturer's instructions. After 24 h cells were lysed, and the lysates were treated with PNGase-F (NEB, P0704L) or water by following the manufacturer's protocol. The samples were resolved by SDS-PAGE and immunoblotted with anti-GFP antibody (described in Immunoblotting section).

## Immunofluorescence analysis

HEK293T or Flp-In™ T-Rex™-293 were fixed with 4% PFA for 10 min on 13 mm coverslips and then washed with PBS three times at room temperature. Next, the cells were permeabilized with 1% TritonX-100 (CAS 9002-93-1) in PBS for 10 min, washed three times with PBS for 5 min and blocked with 1% BSA (Fisher Scientific, 11413164) in PBST (PBS containing 0.1% Tween20 (Fisher Scientific, 10485733)) for 2 h. The cells were then incubated with the corresponding primary antibody diluted in blocking buffer at 4˚C overnight. After washing with PBS for 5 min three times, the corresponding secondary antibodies were added for 1 h at room temperature. Next, DAPI (1:1000 in PBS) was added for 10 min. After washing with PBS for 5 min two times the samples were mounted with antifade mounting medium (Vector Laboratories, H-1700). The samples were imaged on Leica TCS SP8 STED 3X CW 3D microscope using HC PL APO 93x/1.30 motCORR STED WHITE (glycerol, wd 0.3mm) objective. Primary antibodies used for immunofluorescence analysis: rabbit anti-ACBD5, 1:300 (Atlas Antibodies, HPA012145), chicken anti-GFP, 1:10000 (Aves, GFP-1020), rabbit anti-catalase, 1:400 (CST, 12980S), rabbit anti-calnexin, 1:1000 (Abcam, 22595), rabbit anti-ATPIF1, 1:250 (CST, 8528), mouse anti-SDHA, 1:100 (Santa Cruz, 390381), mouse anti-calnexin, 1:500 (Abcam, 112995), mouse anti-HA, 1:400 (Santa Cruz, 2367), mouse anti-catalase 1,6 μg / mL (Biotechne, MAB3398). Secondary antibodies: Alexa Fluor 647 (Invitrogen, A-21235 or CST, 4414S), Alexa Fluor 555 (CST, 4413S or CST, 4409S), Alexa Fluor 488 (Invitrogen, A-11001 or CST, 4412S).

## Colocalization analysis

Colocalization analysis was performed with the Imaris Software (Imaris File Converter 9.5.1) using the Coloc tool of the Surpass system. Images were manually adjusted to separate background from the real signal. Mander's overlap coefficient was used to determine the percentage overlap. For each sample 5–6 fields of view each captured approximately 50 cells were analyzed.

## Immunoblotting

Cells were washed with PBS and lysed in RIPA buffer (CST, 9806). Protein concentration was determined by BCA assay kit (Thermo Scientific, 23227). Protein lysates were supplemented with Laemmli buffer, heated at 95˚C for 5 min and separated by SDS-PAGE. Then the proteins were transferred to a PVDF membrane using Trans-Blot Turbo Mini 0.2 μm PVDF Transfer Pack (BioRad, 1704156). The membranes were blocked in 5% non-fat milk in TBST for 1 h at

room temperature, incubated with appropriate primary antibodies diluted in 1% BSA in TBST for 1 h at RT or overnight at 4°C. Next the membranes were washed three times in TBST followed by 1 h incubation at room temperature with appropriate secondary antibodies diluted in 1% BSA in TBST, washed trice in TBST and imaged for either chemiluminescence or IR signal. Primary antibodies: rabbit anti-GFP 1:2500 (Cell Signaling Technology, 2956), rabbit anti-PEX19 1:1000 (Abcam, 137072), rabbit anti-ACBD5 1:500 (Atlas Antibodies, HPA012145), mouse anti-HA 1:1000 (Cell Signaling Technology, 2367), mouse anti-β-actin, 1:1000 (Cell Signaling Technology, 3700), IRDye 800CW conjugated streptavidin 1:3000 (LI-COR, 926–32230). Secondary antibodies: anti-mouse IgG HRP-linked antibody 1:5000 (Cell Signaling Technology, 7076), anti-rabbit IgG HRP-linked antibody 1:5000 (Cell Signaling Technology, 7074), IRDye 800CW anti-rabbit IgG 1:15000 (LI-COR, 926–32211). The chemiluminescence was developed using ECL substrate for enhanced chemiluminescence (Thermo Scientific, 32106) and the signal was captured by Chemidoc imaging system (Bio-Rad). IR signal was detected using iBright Imaging Systems (Invitrogen).

## Immunoelectron microscopy

WT, PEX3 KO or PEX19 KO HEK293T cells were fixed with paraformaldehyde-lysine-periodate [52], permeabilized with 0.01% saponin (Sigma-Aldrich, S7900) and stained for 1.5 h with anti-GFP antibody (1:500, Sigma, 290), followed by 1 h incubation with 1.4 nm nanogold-conjugated secondary antibody (1:60, RRID: AB_2631182, NY, Nanoprobes, Stony Brook, 2004). Silver enhancement of nanogold particles was performed according to manufacturer's instructions with HQ Silver Kit (Nanoprobes, Stony Brook, NY), then gold toned with 2% Na-Acetate, 0.05% HauCl4, and 0.3% Na2S2O3·5H2O. Post-fixation was done for 1 h on ice with 1% reduced osmium tetroxide in sodium cacodylate buffer pH 7.4, dehydrated by serial washing in ethanol and acetone and finally embedded in Epon (TAAB 812, Aldermaston, UK) for 2 h before 14 h polymerization at 60°C. Sections were prepared with Leica UCT6 microtome, then post-stained with lead citrate and uranyl acetate. Samples were imaged with Jeol JEM-1400 (Jeol Ltd., Tokyo, Japan) operated at 80 kV, with Orius CCD-camera Gatan SC 1000B, bottom mounted (Gatan Inc., Pleasanton, CA). Quantification of the relative labeling index was performed as previously described [53], software used was Microscopy Image Browser [54] to segment and calculate. In case the gold particles were associated with mitochondria or peroxisomal structures, the label was assigned correspondingly to mitochondria or to peroxisomes. If no association with mitochondria or peroxisomes was observed, the gold particle label was assigned to the 'other' category. Therefore, the 'other' category includes all other organelles (except mitochondria and peroxisomes) and cytosol. Nucleus was excluded from this analysis.

## Biotinylation assay

Flp-In™ T-Rex™-293 cells were grown in 6 well-plates and transiently transfected with 250 ng of EGFP- SEC61β-AviTag or 1000 ng of EGFP-YgiM(TA)-AviTag for 24 h and processed for immunoblotting. Membranes were incubated with IRDye 800CW conjugated streptavidin.

## Statistical analysis

All data are presented as mean ± standard deviation (SD). For the statistical analysis with two samples, unpaired two-tailed t-test was performed using Graph Pad Prism 9 Software. All the graphs were prepared with Graph Pad Prism 9 Software.

## Supporting information

**S1 Data.**
(XLSX)

**S1 Raw images.**
(PDF)

## Acknowledgments

We thank Pekka Katajisto for valuable discussion and suggestions, members of Ville Paavilainen laboratory for discussions and insights throughout this project, Maria Vartiainen and Leonardo Souza-Almeida for continuous support.

## Author Contributions

**Data curation:** Tamara Somborac, Svetlana Konovalova.

**Formal analysis:** Tamara Somborac.

**Investigation:** Tamara Somborac, Güleycan Lutfullahoglu Bal, Kaneez Fatima, Helena Vihinen, Anja Paatero, Eija Jokitalo.

**Methodology:** Tamara Somborac, Güleycan Lutfullahoglu Bal.

**Project administration:** Svetlana Konovalova.

**Supervision:** Ville O. Paavilainen, Svetlana Konovalova.

**Visualization:** Tamara Somborac.

**Writing – original draft:** Tamara Somborac.

**Writing – review & editing:** Tamara Somborac, Ville O. Paavilainen, Svetlana Konovalova.

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
