## [Decision Letter · Decision Letter 0]

26 Sep 2023

PONE-D-23-23668The subset of peroxisomal tail-anchored proteins do not reach peroxisomes via ER, instead mitochondria can be involved.PLOS ONE

Dear Dr. Konovalova,

Thank you for submitting your manuscript to PLOS ONE. After careful consideration, we feel that it has merit but does not fully meet PLOS ONE’s publication criteria as it currently stands. Therefore, we invite you to submit a revised version of the manuscript that addresses the points raised during the review process.

We look forward to receiving your revised manuscript.

Kind regards,

Subhadip Mukhopadhyay, PhD

Academic Editor

PLOS ONE

[We gratefully acknowledge grant support provided to Cory Dunn from ERC Starting Grant (grant nr. 637649), Academy of Finland, Finland (grant nr. 331556), Jane ja Aatos Erkon Säätiö, Finland (grant nr. 200057) and Sigrid Jusèlius Foundation, Finland. We thank Pekka Katajisto for valuable discussion and suggestions, members of Ville Paavilainen laboratory for discussions and insights throughout this project, Leonardo Souza-Almeida and Maria Vartiainen for continuous support.]

 [VOP - Academy of Finland (grant nr. 331556, https://www.aka.fi), Sigrid Jusèlius Foundation (https://www.sigridjuselius.fi). The funders had no role in study design, data collection and analysis, decision to publish, or preparation of the manuscript.]

6. We note that Figure 1B, 1D, 2A, 2B, 2D, 3A, 3B, 3C, 4A, 4C and  4D in your submission contain copyrighted images. All PLOS content is published under the Creative Commons Attribution License (CC BY 4.0), which means that the manuscript, images, and Supporting Information files will be freely available online, and any third party is permitted to access, download, copy, distribute, and use these materials in any way, even commercially, with proper attribution. For more information, see our copyright guidelines: http://journals.plos.org/plosone/s/licenses-and-copyright.

A. You may seek permission from the original copyright holder of Figure 1B, 1D, 2A, 2B, 2D, 3A, 3B, 3C, 4A, 4C and  4D to publish the content specifically under the CC BY 4.0 license. 

B. If you are unable to obtain permission from the original copyright holder to publish these figures under the CC BY 4.0 license or if the copyright holder’s requirements are incompatible with the CC BY 4.0 license, please either i) remove the figure or ii) supply a replacement figure that complies with the CC BY 4.0 license. Please check copyright information on all replacement figures and update the figure caption with source information. If applicable, please specify in the figure caption text when a figure is similar but not identical to the original image and is therefore for illustrative purposes only.

Reviewers' comments:

Reviewer's Responses to Questions

**Comments to the Author**

1. Is the manuscript technically sound, and do the data support the conclusions?

Reviewer #1: Partly

Reviewer #2: Yes

2. Has the statistical analysis been performed appropriately and rigorously? 

Reviewer #1: Yes

Reviewer #2: Yes

3. Have the authors made all data underlying the findings in their manuscript fully available?

Reviewer #1: Yes

Reviewer #2: Yes

4. Is the manuscript presented in an intelligible fashion and written in standard English?

Reviewer #1: Yes

Reviewer #2: Yes

5. Review Comments to the Author

Reviewer #1: In this manuscript, Somborac et al. reported a study on the delivery pathway of the tail-anchored (TA) proteins to peroxisomes in mammalian cells. Peroxisome biogenesis is an important research area since its dysfunction can be a reason for both rare and prevalent diseases. From that point of view, the research presented in this manuscript on how peroxisomal membrane proteins (PMPs) are targeted to the membrane of peroxisome would be a nice addition to the understanding of peroxisome biogenesis. However, some sections in the manuscript are confusing and require further explanation. Below are the comments:

1. An important observation of this manuscript is that the peroxisomal TA proteins may be targeted to the mitochondria but not to the ER before their final destination to the peroxisome. However, the authors never mentioned the outcomes or shortcomings (if any) if the TA proteins are transported to the ER prior to their delivery to peroxisomes. How do mitochondria play a beneficial (if any) role in being a targeted organelle for TA protein before its final delivery to the peroxisomal membrane?

2. In figure 1b, authors need to describe the microscope image in detail. What are these cyan and magenta-colored objects mean? The same comment goes for figure 1d. For general readers, authors need to explain the data and point out the observation(s) either in the figure legend or in the results section.

3. In the case of figure 1e, a loading control is necessary. The level of TA proteins is different in different cell lines. Moreover, this glycosylation assay requires proper insertion of the membrane proteins where the glycosylation tag is exposed to the ER lumen. How do the authors confirm that in the absence of peroxisomes, the TA proteins are not aggregated, which of course would fail the glycosylation assay?

4. In figure 2b, the colocalization of YgiM(TA) with both ER and mitochondria contradicts the result of the glycosylation assay that was presented in figure 1e. If PEX3 KO or PEX19 KO cells are showing colocalization of YgiM(TA) with ER, why the glycosylation assay showed no glycosylation bands for YgiM(TA)-OPG protein in those two cell lines? Authors need to justify this observation.

5. In figure 2e, what does ‘other’ stand for? Which organelle(s) were considered for YgiM(TA) colocalization in ‘other’ category?

6. In figure 3, the authors showed that an endogenous peroxisomal TA protein ACBD5 is mostly localized in the mitochondria but not in the ER in PEX19 KO cell lines. Moreover, ACBD5 was barely detectable in PEX3 KO cells. The overall expression of the ACBD5 protein in PEX19 KO cell lines is very poor (figure 3e), which makes the result less convincing. One can argue the colocalization with ER is undetectable because of the expression of the ACBD5 protein in both cell lines.

7. In figure 4, the authors beautifully describe the BirA/AviTag labeling approach. However, in figure 4e, why the biotin/streptavidin labeling efficiency for YgiM(TA)-AviTag protein is more efficient in the IMS fraction in comparison to the PO fraction, where the cell line Flp-In-T-RexTM HEK293T is a wild-type cell line. One would expect efficient labeling of YgiM(TA)-AviTag protein in the PO fraction and a less-efficient labeling in the IMS fraction. The authors need to comment on that.

8. Finally, in the discussion, the authors mentioned a possibility that the delivery of TA proteins to the peroxisomes might not be linear. It may follow a parallel pathway, which leads to a route via mitochondria. Any comment on possible mechanisms that may support this parallel pathway? Is there any information of the regulatory effect behind this linear/dual localization of TA proteins to the peroxisomes?

Overall, this manuscript has potential, however, it requires further explanations of their data. Some control experiments may be required to support their findings.

Reviewer #2: The manuscript entitled “The subset of peroxisomal tail-anchored proteins do not reach peroxisomes via ER,

instead mitochondria can be involved” where authors have demonstrated how mitochondria plays an important role for targeting peroxisomal TA proteins in both presence and absence of peroxisomes. Most importantly, authors have highlighted their findings by several approaches including immunoelectron microscopy. Moreover, authors have also discussed and are curious to investigate mechanism behind the mitochondrial targeting of Peroxisomal tail-anchored proteins.

6. PLOS authors have the option to publish the peer review history of their article (what does this mean?). If published, this will include your full peer review and any attached files.

Reviewer #1: No

Reviewer #2: No

---

## [Author Response · Author response to Decision Letter 0]

17 Oct 2023

The detailed responses to the reviewers’ comments are listed separately as an attachment file.

---

## [Decision Letter · Decision Letter 1]

14 Nov 2023

The subset of peroxisomal tail-anchored proteins do not reach peroxisomes via ER, instead mitochondria can be involved.

PONE-D-23-23668R1

Dear Dr. Konovalova,

We’re pleased to inform you that your manuscript has been judged scientifically suitable for publication and will be formally accepted for publication once it meets all outstanding technical requirements.

Kind regards,

Subhadip Mukhopadhyay, PhD

Academic Editor

PLOS ONE

Additional Editor Comments (optional):

Reviewers' comments:

Reviewer's Responses to Questions

**Comments to the Author**

1. If the authors have adequately addressed your comments raised in a previous round of review and you feel that this manuscript is now acceptable for publication, you may indicate that here to bypass the “Comments to the Author” section, enter your conflict of interest statement in the “Confidential to Editor” section, and submit your "Accept" recommendation.

Reviewer #1: All comments have been addressed

Reviewer #2: (No Response)

2. Is the manuscript technically sound, and do the data support the conclusions?

Reviewer #1: Yes

Reviewer #2: (No Response)

3. Has the statistical analysis been performed appropriately and rigorously? 

Reviewer #1: Yes

Reviewer #2: (No Response)

4. Have the authors made all data underlying the findings in their manuscript fully available?

Reviewer #1: Yes

Reviewer #2: (No Response)

5. Is the manuscript presented in an intelligible fashion and written in standard English?

Reviewer #1: Yes

Reviewer #2: (No Response)

6. Review Comments to the Author

Reviewer #1: In this manuscript, Somborac et al. reported a study on the delivery pathway of the tail-anchored (TA) proteins to peroxisomes in mammalian cells. Overall, the authors did a careful job of editing and revising the content of the manuscript. Enough details were added where needed. The authors have responded all the questions asked by me with reasonable justification. The manuscript reads well and I would recommend publication of this manuscript once it satisfies the journal’s guidelines.

Reviewer #2: The current manuscript “The subset of peroxisomal tail-anchored proteins do not reach peroxisomes via ER, instead mitochondria can be involved” should be accepted.

7. PLOS authors have the option to publish the peer review history of their article (what does this mean?). If published, this will include your full peer review and any attached files.

Reviewer #1: No

Reviewer #2: No

---

## [Editor Report · Acceptance letter]

23 Nov 2023

PONE-D-23-23668R1 

The subset of peroxisomal tail-anchored proteins do not reach peroxisomes via ER, instead mitochondria can be involved. 

Dear Dr. Konovalova:

I'm pleased to inform you that your manuscript has been deemed suitable for publication in PLOS ONE. Congratulations! Your manuscript is now with our production department. 

Kind regards, 

on behalf of

Dr. Subhadip Mukhopadhyay 

Academic Editor

PLOS ONE